# Quasi-Projective Synchronization of Distributed-Order Recurrent Neural Networks

Xiao Liu [1,†], Kelin Li [1,*,†], Qiankun Song [2] and Xujun Yang [2]

1 College of Mathematics and Statistics, Sichuan University of Science & Engineering, Zigong 643000, China; xiaoliu19940130@163.com
2 Department of Mathematics, Chongqing Jiaotong University, Chongqing 400074, China; qiankunsong@163.com (Q.S.); xujunyangcquc@163.com (X.Y.)
* Correspondence: lkl@suse.edu.cn
† These authors contributed equally to this work.

**Abstract:** In this paper, the quasi-projective synchronization of distributed-order recurrent neural networks is investigated. Firstly, based on the definition of the distributed-order derivative and metric space theory, two distributed-order differential inequalities are obtained. Then, by employing the Lyapunov method, Laplace transform, Laplace final value theorem, and some inequality techniques, the quasi-projective synchronization sufficient conditions for distributed-order recurrent neural networks are established in cases of feedback control and hybrid control schemes, respectively. Finally, two numerical examples are given to verify the effectiveness of the theoretical results.

**Keywords:** distributed-order derivative; neural network; quasi-projective synchronization; Laplace final value theorem

## 1. Introduction

Fractional calculus has more merits than classical integer-order calculus in the description of memory and hereditary properties for a variety of materials and processes, and has therefore been of great interest to scholars [1,2]. Many researchers have realized that fractional calculus could be fit for theoretical research and practical application of science and engineering, such as fractal geometry, signal processing, pattern recognition, dynamics of complex dielectric, viscoelastic materials, and automatic control systems [3–8]. As a general rule of fractional calculus, the distributed-order derivative was proposed by Caputo [9]. It is worth noting that a distributed-order derivative is more accurate in describing and explaining some physical phenomena, such as networked structures, the complexity of nonlinear systems, non-homogeneous, multi-scale, and multi-spectral phenomena [10–13]. Therefore, an increasing number of scholars have set out to study distributed-order derivatives. In [14], the authors studied some properties of the viscoelastic rod with a fractional distributed-order derivative and exhibited its potential applications. Moreover, differential equations need to be solved in some cases. In [15,16], the authors presented a numerical method for solving differential equations of distributed-order.

In the past decades, with the development of modern technology, neural networks have been widely applied in various fields, such as signal processing, image recognition, control theory, associative memory, and optimization [17–22]. Nowadays, fractional calculus has been introduced into neural network modelling because it improves the ability of neurons to process information [23]. Fractional order recurrent neural networks, by using sequential information, can enable it to reveal time correlations between nodes that are far away from each other in the data. Thus, the characteristics of memory are endowed to the neural networks. Therefore, it is of great significance to study the dynamic behavior of fractional order recurrent neural networks. In [24], the authors investigated the multistability of fractional-order recurrent neural networks with nonmonotonic activation functions.

In [25], through the Lyapunov method, some conditions for the global Mittag–Leffler stability and synchronization were presented. In [26], several Hopf bifurcation conditions for fractional order recurrent neural networks were presented. The authors pointed out that a fractional-order equation can simulate the activity of neuron oscillation and proposed stability theory of incommensurate fractional-order models. In [27], the authors proposed a fractional-order recurrent neural network sliding mode control scheme for a class of dynamic systems. The proposed RNNFSMC scheme is used for a dynamic model of an active power filter to realize the current harmonic compensation control.

Moreover, synchronization, which has been given full attention by scholars, has a number of potential applications in practice, including cryptography, secure communication, optimization of nonlinear systems performance, etc. [28–35]. Since the two state variables of projection synchronization can synchronize to a scaling factor, this feature has a great application prospect in improving transmission speed by extending binary to M base system in secure communication. In [36], the authors proposed a necessary and sufficient condition for projective synchronization of the chaotic systems in arbitrary dimensions and provided an algorithm to obtain all the solutions of the projective synchronization problem. In [37,38], the authors respectively studied the problem of projective synchronization for fractional order chaotic systems and fractional order delayed neural networks. Authors extend the conclusion of quasi-projection synchronization over real number fields to complex number fields, and give the error bound of quasi-projection synchronization for fractional order recurrent neural networks in [39].

To the best of our knowledge, there are few results on the synchronization of distributed-order recurrent neural networks. Therefore, motivated by the above discussions, in this paper, we investigate the quasi-projective synchronization of distributed-order recurrent neural networks. First, we extend some lemmas that are true at fractional order systems to distributed-order systems. Then, by applying the feedback control, the quasi-projective synchronization of distributed-order recurrent neural networks are obtained. Finally, the quasi-projective synchronization of distributed-order recurrent neural networks are investigated in the case of a hybrid control scheme.

The main contribution of this paper can be summarized in brief as follows. (1) The quasi-projective synchronization of distributed-order recurrent neural networks is investigated for the first time. (2) Based on the definition of distributed-order derivative and metric space theory, two distributed-order differential inequalities are obtained. (3) Several sufficient criteria for quasi-projective synchronization of distributed-order recurrent neural networks are established. (4) The error bound of the model is obtained.

The rest of the paper is structured as follows. In Section 2, the distributed-order recurrent neural networks model and relevant definitions are presented. And we deduced some necessary lemmas and some results that conduce to establish sufficient conditions for the quasi-projective synchronization of distributed-order recurrent neural networks. Then, in Section 3, several sufficient conditions are proposed to ensure the quasi-projective synchronization of such neural network models. Two numerical examples in Section 4 are given to illustrate the effectiveness of the theoretical results. Finally, the paper concludes in Section 5.

*Notations*: In this paper, $D^{\omega(\alpha)}$ denotes $\omega(\alpha)$-order distributed-order fractional derivative operator. $\mathbb{R}$, $\mathbb{R}^n$, and $\mathbb{Z}^+$ respectively indicate the set of real numbers, the $n$-dimensional Euclidean space, and the set of positive integer.

## 2. Model Description and Preliminaries

In this paper, we consider the following distributed-order recurrent neural networks

$$D^{\omega(\alpha)}x_i(t) = -p_i x_i(t) + \sum_{j=1}^{m}\sum_{k=1}^{n} d_{ijk} f_j(x_j(t)) + q_i, \quad i = 1, 2, \cdots, n, \tag{1}$$

where $\omega(\alpha) > 0, \alpha \in (0,1); i,k \in \mathbb{Z}^+ = \{1,2,\cdots,n\}, j \in \mathbb{Z}^+ = \{1,2,\cdots,m\}; x_i(t)$ denotes the state vector; $f_j(x_j(t)): \mathbb{R}^n \rightarrow \mathbb{R}^n$ is the activation functions of the $i$th neuron; $p_i > 0$ denotes the decay rate; $d_{ijk}$ represents the interconnection weight of the $i$th neuron and $j$th neuron; $q_i$ is the constant control input vector.

In order to investigate the synchronization, the response system is depicted as

$$D^{\omega(\alpha)}y_i(t) = -p_i y_i(t) + \sum_{j=1}^{m}\sum_{k=1}^{n} d_{ijk}f_j(y_j(t)) + q_i + u_i(t), \quad i = 1,2,\cdots,n, \tag{2}$$

where $y_i(t)$ denotes the state vector of response system; the rest notations same as in drive system (1). Through feedback control, the controller $u_i(t)$ can be described as

$$u_i(t) = -g_i(y_i(t) - \beta_i x_i(t)). \tag{3}$$

Let $e_i(t) = y_i(t) - \beta_i x_i(t)$, then the error system between the drive system (1) and the response system (2) can be written as

$$D^{\omega(\alpha)}e_i(t) = -(p_i + g_i)e_i(t) + \sum_{j=1}^{m}\sum_{k=1}^{n} d_{ijk}f_j(y_j(t)) - \sum_{j=1}^{m}\sum_{k=1}^{n} d_{ijk}\beta_i f_j(x_j(t)) + (1-\beta_i)q_i. \tag{4}$$

**Hypothesis 1 (H1).** *The function $f_j(x)$ is Lipschitz continuous, i.e., for any $x,y \in \mathbb{R}$, there exists positive constant $F_j$ such that*

$$|f_j(x) - f_j(y)| \leq F_j|x - y|, \quad j = 1,2,\cdots,n. \tag{5}$$

**Hypothesis 2 (H2).** *For any $j \in \mathbb{Z}^+$, there exist positive numbers $H_j$ such that*

$$|f_j| \leq H_j. \tag{6}$$

**Definition 1 ([40]).** *The Caputo fractional derivative of order $\alpha \in \mathbb{R}^+$ is defined as*

$$D^{\alpha}x(t) = \frac{1}{\Gamma(m-\alpha)} \int_0^t (t-\tau)^{m-\alpha-1} x^{(m)}(\tau)d\tau, \tag{7}$$

*where $m - 1 < \alpha < m, m \in \mathbb{Z}^+$.*

**Definition 2 ([41]).** *The distributed-order derivative in the Caputo sense with respect to $\omega(\alpha) > 0$ is defined as*

$$D^{\omega(\alpha)}x(t) = \int_{m-1}^{m} \omega(\alpha)D^{\alpha}x(t)d\alpha, \tag{8}$$

*where $m - 1 < \alpha < m, m \in \mathbb{Z}^+$. In particular, when $\alpha \in (0,1)$, it holds that $D^{\omega(\alpha)}x(t) = \int_0^1 \omega(\alpha)D^{\alpha}x(t)d\alpha$.*

**Definition 3 ([42]).** *The Laplace transform of distributed-order derivative of a function is*

$$\mathcal{L}_{t \rightarrow s}\{D^{\omega(\alpha)}x(t)\} = (\mathcal{L}_{t \rightarrow s}\{x(t)\} - x(0)/s)\int_0^1 \omega(\alpha)s^{\alpha}d\alpha, \tag{9}$$

*where $\mathcal{L}_{t \rightarrow s}\{\cdot\}$ represents the Laplace transform operator.*

**Definition 4 ([39]).** *Systems (1) and (2) can be said to be quasi-projectively synchronized, if there exists a small error bound $\epsilon > 0$ such that*

$$\lim_{t \rightarrow \infty} |y_i(t) - \beta_i x_i(t)| \leq \epsilon, \tag{10}$$

*where $\beta_i \in \mathbb{R}$ is the projective coefficient and $\beta_i \neq 0$.*

**Lemma 1.** *Suppose function $h(t) \in \mathbb{R}$ is continuous and differentiable on $t \in [t_0, \infty)$, it has*

$$D^{\omega(\alpha)} h(t)^2 \leq 2h(t) D^{\omega(\alpha)} h(t), \tag{11}$$

*where $\omega(\alpha) > 0$, $\alpha \in (0, 1)$.*

**Proof.** From Definitions 1 and 2, we have

$$
\begin{aligned}
D^{\omega(\alpha)} h(t)^2 &= \int_{t_0}^{t} \omega(\alpha) \frac{1}{\Gamma(1-\alpha)} \int_{t_0}^{t} \frac{2(h(\tau)h'(\tau))}{(t-\tau)^\alpha} d\tau d\alpha \\
&= \int_{t_0}^{t} \omega(\alpha) \frac{2}{\Gamma(1-\alpha)} \int_{t_0}^{t} \frac{(h(\tau) - h(t) + h(t))h'(\tau)}{(t-\tau)^\alpha} d\tau d\alpha \\
&= \int_{t_0}^{t} \omega(\alpha) \left[ \frac{2}{\Gamma(1-\alpha)} \int_{t_0}^{t} \frac{(h(\tau) - h(t))h'(\tau)}{(t-\tau)^\alpha} d\tau + 2h(t) D^\alpha h(t) \right] d\alpha \\
&= \int_{t_0}^{t} \omega(\alpha) \left[ \frac{2}{\Gamma(1-\alpha)} \int_{t_0}^{t} \frac{\phi(\tau)\phi'(\tau)}{(t-\tau)^\alpha} d\tau + 2h(t) D^\alpha h(t) \right] d\alpha,
\end{aligned}
\tag{12}
$$

where

$$
\begin{aligned}
\int_{t_0}^{t} \frac{\phi(\tau)\phi'(\tau)}{(t-\tau)^\alpha} d\tau &= \frac{\phi^2(\tau)}{(t-\tau)^\alpha} \Big|_{t_0}^{t} - \int_{t_0}^{t} \phi(\tau) d \frac{\phi(\tau)}{(t-\tau)^\alpha} \\
&= \lim_{\tau \to t} \frac{\phi^2(\tau)}{(t-\tau)^\alpha} - \frac{\phi^2(t_0)}{(t-t_0)^\alpha} - \int_{t_0}^{t} \frac{\phi(\tau)\phi'(\tau)}{(t-\tau)^\alpha} d\tau - \alpha \int_{t_0}^{t} \frac{\phi^2(\tau)}{(t-\tau)^{\alpha+1}} d\tau.
\end{aligned}
\tag{13}
$$

By transposing, it can obtain

$$2 \int_{t_0}^{t} \frac{\phi(\tau)\phi'(\tau)}{(t-\tau)^\alpha} d\tau = \lim_{\tau \to t} \frac{\phi^2(\tau)}{(t-\tau)^\alpha} - \frac{\phi^2(t_0)}{(t-t_0)^\alpha} - \alpha \int_{t_0}^{t} \frac{\phi^2(\tau)}{(t-\tau)^{\alpha+1}} d\tau, \tag{14}$$

then, we have

$$
\begin{aligned}
\int_{t_0}^{t} \frac{\phi(\tau)\phi'(\tau)}{(t-\tau)^\alpha} d\tau &= \frac{1}{2} \left[ \lim_{\tau \to t} \frac{\phi^2(\tau)}{(t-\tau)^\alpha} - \frac{\phi^2(t_0)}{(t-t_0)^\alpha} - \alpha \int_{t_0}^{t} \frac{\phi^2(\tau)}{(t-\tau)^{\alpha+1}} d\tau \right] \\
&= \frac{1}{2} \left[ \lim_{\tau \to t} \frac{2\phi(\tau)\phi'(\tau)(t-\tau)^{1-\alpha}}{\alpha} - \frac{\phi^2(t_0)}{(t-t_0)^\alpha} - \alpha \int_{t_0}^{t} \frac{\phi^2(\tau)}{(t-\tau)^{\alpha+1}} d\tau \right] \\
&\leq 0.
\end{aligned}
\tag{15}
$$

Submitting (15) into (12), we have

$$D^{\omega(\alpha)} h(t)^2 \leq 2h(t) D^{\omega(\alpha)} h(t). \tag{16}$$

$\square$

For more details of the proof process, one can refer to the proof of Theorem 2 in [43].

**Lemma 2.** *If $x(t) \in C^1([0, +\infty], R)$ is a continuously differentiable function, for any $\omega(\alpha) > 0$ and $\alpha \in (0, 1)$, the following inequality holds almost everywhere*

$$D^{\omega(\alpha)} |x(t)| \leq sgn(x(t)) D^{\omega(\alpha)} x(t). \tag{17}$$

**Proof.** It is easy to obtain that $|x(t)|$ is differentiable except several points, which belong to the set $\Omega = \{t \,|\, x(t) = 0, x'(t) \neq 0\}$. In addition, according to measure theory, we assert that $\Omega$ is a countable set and its measure is 0.

Afterwards, we prove that inequation (17) holds except at the set $\Omega$. Without loss of generality, the trajectory of $|x(t)|$ can be depicted as Figure 1. The solid line indicating

$|x(t)|$ is divided into two parts by $S_3 \in \Omega$. Suppose that there exists only one extreme point as point $S_2$ and every part is divided into two pieces like $R_1, R_2$ in part A. The dash line is the trajectory of $-|x(t)|$ which may be the one of $x(t)$ in any part. $(x_s, y_s)$ denotes the coordinate of point $S$ in Figure 1. When $t \notin \Omega$, we have

$$D^{\omega(\alpha)}|x(t)| = \int_0^1 \omega(\alpha) \frac{1}{\Gamma(1-\alpha)} \int_0^t \frac{|x(t)|'}{(t-\tau)^\alpha} d\tau d\alpha$$
$$= \int_0^1 \omega(\alpha) \frac{1}{\Gamma(1-\alpha)} \left( \int_0^{x_{S_3}} + \int_{x_{S_3}}^t \frac{|x(\tau)|'}{(t-\tau)^\alpha} d\tau \right) d\alpha, \tag{18}$$

and

$$sgn(x(t)) D^{\omega(\alpha)} x(t) = sgn(x(t)) \int_0^1 \omega(\alpha) \frac{1}{\Gamma(1-\alpha)} \left( \int_0^{x_{S_3}} + \int_{x_{S_3}}^t \frac{x(\tau)'}{(t-\tau)^\alpha} d\tau \right) d\alpha. \tag{19}$$

In Part A, if $sgn(x(t))x(\tau) = |x(\tau)|$, then $\int_0^{x_{S_3}} \frac{|x(t)|'}{(t-\tau)^\alpha} d\tau = \int_0^{x_{S_3}} \frac{sgn(x(t))x(\tau)'}{(t-\tau)^\alpha} d\tau$; lif $sgn(x(t))$ $x(\tau) = -|x(\tau)|$, according to the properties of integrals and derivatives, we have

$$\int_0^{x_{S_3}} \frac{|x(t)|'}{(t-\tau)^\alpha} d\tau = \int_0^{x_{S_2}} \frac{|x(t)|'}{(t-\tau)^\alpha} d\tau + \int_{x_{S_2}}^{x_{S_3}} \frac{|x(t)|'}{(t-\tau)^\alpha} d\tau$$
$$< \frac{1}{(t-x_{S_2})^\alpha} \int_0^{x_{S_2}} |x(t)|' d\tau + \frac{1}{(t-x_{S_2})^\alpha} \int_{x_{S_2}}^{x_{S_3}} |x(t)|' d\tau \tag{20}$$
$$= \frac{-y_{S_1}}{(t-x_{S_2})^\alpha} \leq 0.$$

Due to $sgn(x(t))x(\tau) = -|x(\tau)|$, then we have

$$\int_0^{x_{S_3}} \frac{|x(t)|'}{(t-\tau)^\alpha} d\tau < \int_0^{x_{S_3}} \frac{sgn(x(t))x(\tau)'}{(t-\tau)^\alpha} d\tau. \tag{21}$$

In Part B, whether $x(t) > 0$ or $x(t) < 0$, $sgn(x(t))x(\tau) = |x(\tau)|$ holds, thus

$$\int_{x_{S_5}}^t \frac{|x(t)|'}{(t-\tau)^\alpha} d\tau = \int_{x_{S_5}}^t \frac{sgn(x(t))x(\tau)'}{(t-\tau)^\alpha} d\tau. \tag{22}$$

To sum up, inequation (17) holds almost everywhere. For the case where multiple extreme points exist, in [44] the authors prove that the inequation (17) is still valid. □

**Remark 1.** *It is noted that a distributed-order derivative is a generalization of a fractional derivative. In Lemmas 1 and 2, if the order of the distributed-order derivative changes from $\omega(\alpha)$ to $\alpha$, the above two distributed-order inequalities become fractional inequalities, and their conclusions are proved in references [43,44].*

**Lemma 3** ([45]). *Let $\lambda_1 > 0$, $\lambda_2 > 0$, $\lambda_3 > 1$, $\lambda_4 > 1$ and $\frac{1}{\lambda_3} + \frac{1}{\lambda_4} = 1$. Then, for any $\varrho > 0$, it holds that*

$$\lambda_1 \lambda_2 \leq \frac{1}{\lambda_3} (\lambda_1 \varrho)^{\lambda_3} + \frac{1}{\lambda_4} \left( \lambda_2 \frac{1}{\varrho} \right)^{\lambda_4}. \tag{23}$$

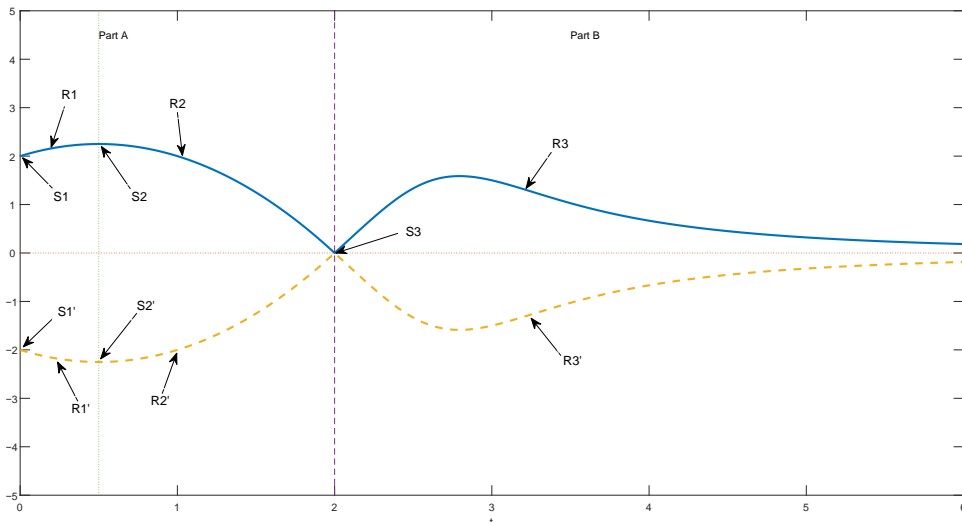

**Figure 1.** The $|x(t)|$'s trajectory.

## 3. Main Results

**Theorem 1.** *Under assumptions (H1) and (H2), if the following inequalities hold*

$$\lambda_1 = \sum_{i=1}^{n} \sum_{j=1}^{m} \sum_{k=1}^{n} \left[ 2(p_i + g_i) - 2d_{ijk}^2 - \frac{\xi_j}{\xi_i} F_i^2 - 1 \right] > 0, \tag{24}$$

$$\lambda_2 = \sum_{i=1}^{n} \xi_i \left[ \sum_{j=1}^{m} \sum_{k=1}^{n} H_j^2 (1 + \beta_i^2 - 2\beta_i) + (1 - \beta_i)^2 q_i^2 \right] > 0, \tag{25}$$

$$\mathcal{L}^{-1} \left\{ \frac{1}{\int_0^1 \omega(\alpha) s^\alpha d\alpha + \lambda_1} \right\} \geq 0, \tag{26}$$

*and the roots of $\int_0^1 \omega(\alpha) s^\alpha d\alpha + \lambda_1$ are in the open left-half complex plane, then systems (1) and (2) can achieve quasi-projectively synchronized under the controller (3).*

**Proof.** Consider the Lyapunov function candidate as follows

$$V(t) = \sum_{i=1}^{n} \xi_i e_i^2(t), \tag{27}$$

where $\xi_i$ is positive constant.

According to Lemma 1 , it shows that

$$
\begin{aligned}
D^{\omega(\alpha)} V(t) &\leq \sum_{i=1}^{n} 2\xi_i e_i(t) D^{\omega(\alpha)} e_i(t) \\
&= 2 \sum_{i=1}^{n} \xi_i e_i(t) \left[ -(p_i + g_i) e_i(t) + \sum_{j=1}^{m} \sum_{k=1}^{n} d_{ijk} \left( f_j(y_j(t)) - f_j(\beta_j x_j(t)) \right) \right. \\
&\quad \left. + \sum_{j=1}^{m} \sum_{k=1}^{n} d_{ijk} \left( f_j(\beta_j x_j(t)) - \beta_i f_j(x_j(t)) \right) + (1 - \beta_i) q_i \right] \\
&= \sum_{i=1}^{n} \xi_i \left[ -2(p_i + g_i) e_i^2(t) \right] + \sum_{i=1}^{n} \xi_i \left[ \sum_{j=1}^{m} \sum_{k=1}^{n} 2e_i(t) d_{ijk} \left( f_j(y_j(t)) - f_j(\beta_j x_j(t)) \right) \right)
\end{aligned}
$$

$$
+ \sum_{j=1}^{m} \sum_{k=1}^{n} 2e_i(t)d_{ijk} \Big( f_j(\beta_j x_j(t)) - \beta_i f_j(x_j(t)) \Big) \Bigg] + \sum_{i=1}^{n} \xi_i \Big[ 2e_i(t)(1 - \beta_i)q_i \Big]
$$

$$
\leq \sum_{i=1}^{n} \xi_i e_i^2(t) \Big[ -2(p_i + g_i) \Big] + \sum_{i=1}^{n} \xi_i \Big[ d_{ijk}^2 e_i^2(t) + \Big( f_j(y_j(t)) - f_j(\beta_j x_j(t)) \Big)^2 \Big]
$$

$$
+ \sum_{i=1}^{n} \xi_i \Big[ d_{ijk}^2 e_i^2(t) + \Big( f_j(\beta_j x_j(t)) - \beta_i f_j(x_j(t)) \Big)^2 \Big] + \sum_{i=1}^{n} \Big[ \xi_i e_i^2(t) + \xi_i(1 - \beta_i)^2 q_i^2 \Big]
$$

$$
\leq \sum_{i=1}^{n} \xi_i e_i^2(t) \Big[ -2(p_i + g_i) \Big] + \sum_{i=1}^{n} \xi_i e_i^2(t) \Big[ \sum_{j=1}^{m} \sum_{k=1}^{n} \Big( d_{ijk}^2 + \frac{\xi_j}{\xi_i} F_i^2 \Big) \Big]
$$
$$ \tag{28} $$

$$
+ \sum_{i=1}^{n} \xi_i \Big[ \sum_{j=1}^{m} \sum_{k=1}^{n} d_{ijk}^2 e_i^2(t) + H_j^2(1 + \beta_i^2 - 2\beta_i) \Big] + \sum_{i=1}^{n} \Big[ \xi_i e_i^2(t) + \xi_i(1 - \beta_i)^2 q_i^2 \Big]
$$

$$
= - \sum_{i=1}^{n} \sum_{j=1}^{m} \sum_{k=1}^{n} \xi_i e_i^2(t) \Big[ 2(p_i + g_i) - 2d_{ijk}^2 - \frac{\xi_j}{\xi_i} F_i^2 - 1 \Big]
$$

$$
+ \sum_{i=1}^{n} \xi_i \Big[ \sum_{j=1}^{m} \sum_{k=1}^{n} H_j^2(1 + \beta_i^2 - 2\beta_i) + (1 - \beta_i)^2 q_i^2 \Big]
$$

$$
\leq -\lambda_1 V(t) + \lambda_2,
$$

where

$$
\lambda_1 = \sum_{i=1}^{n} \sum_{j=1}^{m} \sum_{k=1}^{n} \Big[ 2(p_i + g_i) - 2d_{ijk}^2 - \frac{\xi_j}{\xi_i} F_i^2 - 1 \Big]
$$

and

$$
\lambda_2 = \sum_{i=1}^{n} \xi_i \Big[ \sum_{j=1}^{m} \sum_{k=1}^{n} H_j^2(1 + \beta_i^2 - 2\beta_i) + (1 - \beta_i)^2 q_i^2 \Big].
$$

Since $\lambda_1 > 0, \lambda_2 > 0$, hence, we can find $n(t) \geq$ such that

$$
D^{\omega(\alpha)} V(t) = -\lambda_1 V(t) + \lambda_2 - n(t). \tag{29}
$$

Taking the Laplace transform from both sides of (29), one has

$$
\Big( V(s) - \frac{V(0)}{s} \Big) \int_0^1 \omega(\alpha)s^\alpha d\alpha + N(s) = -\lambda_1 V(s) + \frac{\lambda_2}{s}. \tag{30}
$$

It follows from (30) that

$$
V(s) = \frac{\frac{\lambda_2 + V(0) \int_0^1 \omega(\alpha)s^\alpha d\alpha}{s} - N(s)}{\int_0^1 \omega(\alpha)s^\alpha d\alpha + \lambda_1}, \tag{31}
$$

then

$$
\begin{aligned}
V(t) &= \mathcal{L}^{-1} \left\{ \frac{\lambda_2 + V(0) \int_0^1 \omega(\alpha)s^\alpha d\alpha}{s \Big( \int_0^1 \omega(\alpha)s^\alpha d\alpha + \lambda_1 \Big)} \right\} - \mathcal{L}^{-1} \left\{ \frac{N(s)}{\int_0^1 \omega(\alpha)s^\alpha d\alpha + \lambda_1} \right\} \\
&= \mathcal{L}^{-1} \left\{ \frac{\lambda_2 + V(0) \int_0^1 \omega(\alpha)s^\alpha d\alpha}{s \Big( \int_0^1 \omega(\alpha)s^\alpha d\alpha + \lambda_1 \Big)} \right\} - N(t) * g(t) \\
&= \mathcal{L}^{-1} \left\{ \frac{\lambda_2 + V(0) \int_0^1 \omega(\alpha)s^\alpha d\alpha}{s \Big( \int_0^1 \omega(\alpha)s^\alpha d\alpha + \lambda_1 \Big)} \right\} - \int_0^t N(\eta)g(t - \eta)d\eta,
\end{aligned} \tag{32}
$$

where $g(t) = \mathcal{L}^{-1}\left\{\frac{\lambda_2 + V(0)\int_0^1 \omega(\alpha)s^\alpha d\alpha}{s\left(\int_0^1 \omega(\alpha)s^\alpha d\alpha + \lambda_1\right)}\right\}$. Since $g(t) \geq 0, N(t) \geq 0$, then

$$V(t) \leq \mathcal{L}^{-1}\left\{\frac{\lambda_2 + V(0)\int_0^1 \omega(\alpha)s^\alpha d\alpha}{s\left(\int_0^1 \omega(\alpha)s^\alpha d\alpha + \lambda_1\right)}\right\}. \tag{33}$$

In light of Laplace final value theorem, we have

$$\begin{aligned}
\lim_{t\to+\infty} V(t) &\leq \lim_{t\to+\infty} \mathcal{L}^{-1}\left\{\frac{\lambda_2 + V(0)\int_0^1 \omega(\alpha)s^\alpha d\alpha}{s\left(\int_0^1 \omega(\alpha)s^\alpha d\alpha + \lambda_1\right)}\right\} \\
&= \lim_{s\to 0} \frac{\lambda_2 + V(0)\int_0^1 \omega(\alpha)s^\alpha d\alpha}{\int_0^1 \omega(\alpha)s^\alpha d\alpha + \lambda_1} \\
&= \frac{\lambda_2}{\lambda_1}.
\end{aligned} \tag{34}$$

Furthermore, one has

$$\min_{1\leq i\leq n} \xi_i \lim_{t\to+\infty} \sum_{i=1}^n e_i^2(t) \leq \lim_{t\to+\infty} \sum_{i=1}^n \xi_i e_i^2(t), \tag{35}$$

which means that

$$\lim_{t\to+\infty} \|e(t)\|_2 \leq \sqrt{\frac{\lambda_2}{\lambda_1 \min_{1\leq i\leq n} \xi_i}}. \tag{36}$$

Thus, in accordance with Definition 4, system (1) and system (2) can be said to be quasi-projectively synchronized. □

**Theorem 2.** *Under assumptions (H1) and (H2), if the following inequalities hold*

$$\lambda_3 = \sum_{i=1}^n \sum_{j=1}^m \sum_{k=1}^n \left[2(p_i + g_i) - d_{ijk} - d_{jik}F_i^2 - d_{ijk}\mu - 1\right] > 0, \tag{37}$$

$$\lambda_4 = \sum_{i=1}^n \sum_{j=1}^m \sum_{k=1}^n \left[\xi_i d_{ijk}\frac{2}{\mu}H_j^2(1 + \beta_i^2) + \xi_i(1 - \beta_i)^2 q_i^2\right] > 0, \tag{38}$$

$$\mathcal{L}^{-1}\left\{\frac{1}{\int_0^1 \omega(\alpha)s^\alpha d\alpha + \lambda_3}\right\} \geq 0, \tag{39}$$

*and the roots of $\int_0^1 \omega(\alpha)s^\alpha d\alpha + \lambda_3$ are in the open left-half complex plane, then systems (1) and (2) can achieve quasi-projectively synchronization under the controller (3).*

**Proof.** Consider the Lyapunov function candidate as follows

$$V(t) = \sum_{i=1}^n \xi_i e_i^2(t), \tag{40}$$

where $\xi_i$ is positive constant.

According to Lemmas 1 and 3, for any positive constant $\mu$, it shows that

$$
\begin{aligned}
D^{\omega(\alpha)}V(t) &\leq \xi_i e_i(t) D^{\omega(\alpha)} e_i(t) \\
&= 2\sum_{i=1}^n \xi_i e_i(t)\Bigg[ -(p_i+g_i)e_i(t) + \sum_{j=1}^m \sum_{k=1}^n d_{ijk}\Big(f_j(y_j(t)) - f_j(\beta_j x_j(t))\Big) \\
&\quad + \sum_{j=1}^m \sum_{k=1}^n d_{ijk}\Big(f_j(\beta_j x_j(t)) - \beta_i f_j(x_j(t))\Big) + (1-\beta_i)q_i \Bigg] \\
&= \sum_{i=1}^n \xi_i\Bigg[ -2(p_i+g_i)e_i^2(t)\Bigg] + \sum_{i=1}^n \xi_i\Bigg[ \sum_{j=1}^m \sum_{k=1}^n 2e_i(t)d_{ijk}\Big(f_j(y_j(t)) - f_j(\beta_j x_j(t))\Big) \\
&\quad + \sum_{j=1}^m \sum_{k=1}^n 2e_i(t)d_{ijk}\Big(f_j(\beta_j x_j(t)) - \beta_i f_j(x_j(t))\Big)\Bigg] + \sum_{i=1}^n \xi_i\Bigg[ 2e_i(t)(1-\beta_i)q_i\Bigg] \\
&\leq \sum_{i=1}^n \xi_i e_i^2(t)\Bigg[ -2(p_i+g_i)\Bigg] + \sum_{i=1}^n \xi_i\Bigg[ \sum_{j=1}^m \sum_{k=1}^n 2d_{ijk}e_i(t)e_j(t)F_j\Bigg] \\
&\quad + \sum_{i=1}^n \sum_{j=1}^m \sum_{k=1}^n \xi_i d_{ijk}\Bigg[ \mu e_i^2(t) + \frac{1}{\mu}\Big(f_j(\beta_j x_j(t)) - \beta_i f_j(x_j(t))\Big)^2\Bigg] \\
&\quad + \sum_{i=1}^n \Bigg[ \xi_i e_i^2(t) + \xi_i(1-\beta_i)^2 q_i^2\Bigg] \\
&\leq \sum_{i=1}^n \xi_i e_i^2(t)\Bigg[ -2(p_i+g_i)\Bigg] + \sum_{i=1}^n \sum_{j=1}^m \sum_{k=1}^n \xi_i e_i^2(t)\Bigg[ d_{ijk} + d_{jik}F_i^2\Bigg] \\
&\quad + \sum_{i=1}^n \sum_{j=1}^m \sum_{k=1}^n \xi_i d_{ijk}\Bigg[ \mu e_i^2 + \frac{2}{\mu}\Big(f_j(\beta_j x_j(t))^2 + \beta_i^2 f_j(x_j(t))^2\Big)\Bigg] + \sum_{i=1}^n \Bigg[ \xi_i e_i^2(t) \\
&\quad + \xi_i(1-\beta_i)^2 q_i^2\Bigg] \\
&\leq \sum_{i=1}^n \xi_i e_i^2(t)\Bigg[ -2(p_i+g_i)\Bigg] + \sum_{i=1}^n \sum_{j=1}^m \sum_{k=1}^n \xi_i e_i^2(t)\Bigg[ d_{ijk} + d_{jik}F_i^2 + d_{ijk}\mu\Bigg] \\
&\quad + \sum_{i=1}^n \sum_{j=1}^m \sum_{k=1}^n \xi_i d_{ijk}\Bigg[ \frac{2}{\mu}H_j^2(1+\beta_i^2)\Bigg] + \sum_{i=1}^n \Bigg[ \xi_i e_i^2(t) + \xi_i(1-\beta_i)^2 q_i^2\Bigg] \\
&= -\lambda_3 V(t) + \lambda_4,
\end{aligned}
\tag{41}
$$

where $\lambda_3 = \sum_{i=1}^n \sum_{j=1}^m \sum_{k=1}^n \Big[ 2(p_i+g_i) - d_{ijk} - d_{jik}F_i^2 - d_{ijk}\mu - 1\Big]$ and $\lambda_4 = \sum_{i=1}^n \sum_{j=1}^m$

$\sum_{k=1}^n \Big[ \xi_i d_{ijk}\frac{2}{\mu}H_j^2(1+\beta_i^2) + \xi_i(1-\beta_i)^2 q_i^2\Big]$.

Same as the proof of Theorem 1, by applying the Laplace transform and final value theorem, one has

$$
\lim_{t\to+\infty} \|e(t)\|_2 \leq \sqrt{\frac{\lambda_4}{\lambda_3 \min_{1\leq i\leq n} \xi_i}}.
\tag{42}
$$

Thus, in accordance with Definition 4, systems (1) and (2) can be said to be quasi-projectively synchronized. □

Depending on different proof methods, we obtain Theorems 1 and 2, which have different conditions. However, they both guarantee that systems (1) and (2) can achieve quasi-projectively synchronization.

Next, we consider the following hybrid control scheme, the controller $\hat{u}_i(t)$ can be described as

$$\hat{u}_i(t) = -g_i(y_i(t) - \beta_i x_i(t)) - \sum_{j=1}^{m}\sum_{k=1}^{n} d_{ijk} f_j(\beta_j x_j(t)) + \sum_{j=1}^{m}\sum_{k=1}^{n} d_{ijk}\beta_i f_j(x_j(t)) - (1-\beta_i)q_i. \quad (43)$$

**Theorem 3.** *Under assumptions (H1) and (H2), if the following inequality holds*

$$\lambda_5 = \sum_{i=1}^{n}\sum_{j=1}^{m}\sum_{k=1}^{n}\left[2(p_i + g_i) - \mu d_{ijk}F_j - \frac{1}{\mu}d_{jik}F_i\right] > 0, \quad (44)$$

$$\mathcal{L}^{-1}\left\{\frac{1}{\int_0^1 \omega(\alpha)s^\alpha d\alpha + \lambda_5}\right\} \geq 0, \quad (45)$$

*and the roots of $\int_0^1 \omega(\alpha)s^\alpha d\alpha + \lambda_5$ are in the open left-half complex plane, then systems (1) and (2) can achieve quasi-projectively synchronization under the controller (43).*

**Proof.** Consider the Lyapunov function candidate as follows

$$V(t) = \sum_{i=1}^{n}\frac{1}{2}|e_i(t)|^2, \quad (46)$$

where $\xi_i$ is positive constant.

According to Lemmas 1 and 2, for any positive constant $\mu$, it shows that

$$
\begin{aligned}
D^{\omega(\alpha)}V(t) &= \sum_{i=1}^{n}\frac{1}{2}D^{\omega(\alpha)}|e_i(t)|^2 \\
&\leq \sum_{i=1}^{n}|e_i(t)|D^{\omega(\alpha)}|e_i(t)| \\
&\leq \sum_{i=1}^{n}|e_i(t)|sgn(e_i(t))D^{\omega(\alpha)}e_i(t) \\
&= \sum_{i=1}^{n}|e_i(t)|sgn(e_i(t))\left[-(p_i+g_i)e_i(t) + \sum_{j=1}^{m}\sum_{k=1}^{n}d_{ijk}\left(f_j(y_j(t)) - f_j(\beta_j x_j(t))\right)\right] \\
&\leq \sum_{i=1}^{n}|e_i(t)|\left[-(p_i+g_i)|e_i(t)| + \sum_{j=1}^{m}\sum_{k=1}^{n}d_{ijk}F_j|e_j(t)|\right] \\
&= \sum_{i=1}^{n}\left[-(p_i+g_i)|e_i(t)|^2 + \sum_{j=1}^{m}\sum_{k=1}^{n}d_{ijk}F_j|e_j(t)||e_i(t)|\right].
\end{aligned}
\quad (47)
$$

By applying Lemma 3, we can obtain that

$$
\begin{aligned}
D^{\omega(\alpha)}V(t) &\leq \sum_{i=1}^{n}\left[-(p_i+g_i)|e_i(t)|^2 + \sum_{j=1}^{m}\sum_{k=1}^{n}d_{ijk}F_j\left(\frac{\mu}{2}|e_i(t)|^2 + \frac{1}{2\mu}|e_j(t)|^2\right)\right] \\
&= \sum_{i=1}^{n}\left[-(p_i+g_i)|e_i(t)|^2 + \frac{\mu}{2}\sum_{j=1}^{m}\sum_{k=1}^{n}d_{ijk}F_j|e_i(t)|^2 + \frac{1}{2\mu}\sum_{j=1}^{m}\sum_{k=1}^{n}d_{jik}F_i|e_i(t)|^2\right] \\
&= -\lambda_5 V(t),
\end{aligned}
\quad (48)
$$

where $\lambda_5 = \sum_{i=1}^{n}\sum_{j=1}^{m}\sum_{k=1}^{n}\left[2(p_i+g_i) - \mu d_{ijk}F_j - \frac{1}{\mu}d_{jik}F_i\right]$.

Same as the proof of Theorem (1), by applying the Laplace transform and final value theorem, one has

$$\lim_{t \to +\infty} \|e(t)\|_2 \le \sqrt{\frac{1}{\lambda_5 \min_{1 \le i \le n} \xi_i}}. \tag{49}$$

Thus, in accordance with Definition 4, system (1) and system (2) can be said to be quasi-projectively synchronized. □

## 4. Numerical Simulation Examples

By applying the predictor−corrector scheme [46], two numerical examples are given to demonstrate the theoretical results in this section.

**Example 1.** *Consider the following distributed-order system*

$$\begin{cases} D^{\omega(\alpha)}x_1(t) = -p_1 x_1(t) + \sum_{j=1}^{2}\sum_{k=1}^{2} d_{1jk} f_j(x_j(t)) + q_1, \\ \\ D^{\omega(\alpha)}x_2(t) = -p_2 x_2(t) + \sum_{j=1}^{2}\sum_{k=1}^{2} d_{2jk} f_j(x_j(t)) + q_2, \end{cases} \tag{50}$$

*where* $\omega(\alpha) = \tau_1 \delta(\alpha - \alpha_1) + \tau_2 \delta(\alpha - \alpha_2)$, $\alpha_1 = \frac{2}{3}, \alpha_2 = \frac{1}{3}$, $\delta(\cdot)$ *denotes Dirac function;* $f_j(x_j(x)) = \tanh(x)$, $j = 1, 2; q_i = 1$, $i = 1, 2$ *and*

$$P = diag(p_1, p_2) = \begin{pmatrix} 1.03125 & 0 \\ 0 & 1 \end{pmatrix},$$

$$D_1 = (d_{ij1})_{2 \times 2} = \begin{pmatrix} 1 & 0 \\ 0 & 1 \end{pmatrix},$$

$$D_2 = (d_{ij2})_{2 \times 2} = \begin{pmatrix} 1 & 0 \\ 1 & 0 \end{pmatrix},$$

*The response system is depicted as*

$$\begin{cases} D^{\omega(\alpha)}y_1(t) = -p_1 y_1(t) + \sum_{j=1}^{2}\sum_{k=1}^{2} d_{1jk} f_j(y_j(t)) + q_1 + u_1(t), \\ \\ D^{\omega(\alpha)}y_2(t) = -p_2 y_2(t) + \sum_{j=1}^{2}\sum_{k=1}^{2} d_{2jk} f_j(y_j(t)) + q_2 + u_2(t), \end{cases} \tag{51}$$

*where* $f_j(y_j(x)) = \tanh(y)$, $j = 1, 2$ *and* $\omega(\alpha), P, D_i, q_i$, $i = 1, 2$ *are the same as the drive system* (50).

*Take* $\beta_1 = 1.5, \beta_2 = 1, g_1 = g_2 = 2, \tau_1 = 4, \tau_2 = 1$, *by calculation, we have* $\sum_{i=1}^{n} \sum_{j=1}^{m} \sum_{k=1}^{n}$

$\left[2(p_i + g_i) - 2d_{ijk}^2 - \frac{\xi_j}{\xi_i} F_i^2 - 1\right] = 0.0625 > 0$ *and the error bounded* $\sqrt{\frac{\lambda_2}{\lambda_1 \min_{1 \le i \le n} \xi_i}} \approx 2.83$.

*Moreover, one has*

$$\int_0^1 \omega(\alpha)s^\alpha d\alpha + \lambda_1 = 4s^{\frac{2}{3}} + s^{\frac{1}{3}} + \frac{1}{16} = 0, \tag{52}$$

*therefore* $s = \frac{-1}{512}$, *which are located in the open left-half complex plane. In addition,*

$$\mathcal{L}^{-1}\left\{\frac{1}{\int_0^1 \omega(\alpha)s^\alpha d\alpha + \lambda_1}\right\} = \mathcal{L}^{-1}\left\{\frac{1}{(2s^{\frac{1}{3}} + \frac{1}{4})^2}\right\}$$

$$= \frac{1}{4}\mathcal{L}^{-1}\left\{\frac{1}{(s^{\frac{1}{3}} + \frac{1}{8})^2}\right\} \tag{53}$$

$$= \frac{1}{4}\int_0^t \eta^{\frac{-2}{3}} E_{\frac{1}{3},\frac{1}{3}}\left(\frac{-1}{8}\eta^{\frac{1}{3}}\right)(t-\eta)^{\frac{-2}{3}} E_{\frac{1}{3},\frac{1}{3}}\left(\frac{-1}{8}(t-\eta)^{\frac{1}{3}}\right) d\eta$$

$$\geq 0,$$

*which means that all conditions in Theorem 1 hold. Hence, it follows from Theorem 1 that systems (50) and (51) can achieve quasi-projectively synchronization. Figure 2 is shows the numerical simulation result.*

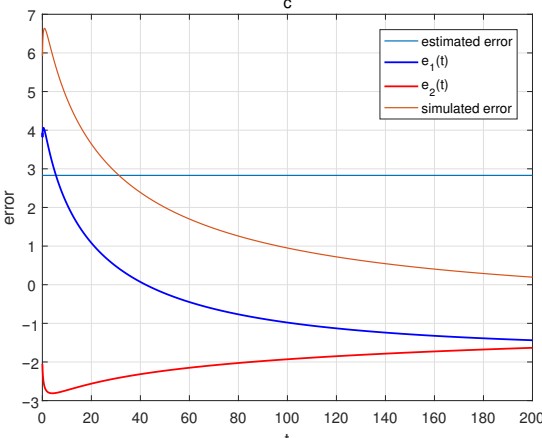

**Figure 2.** The state of error system when $g_i = 2, \beta_1 = 1.5, \beta_2 = 1$.

**Example 2.** *Consider the following distributed-order system*

$$\begin{cases} D^{\omega(\alpha)}x_1(t) = -p_1 x_1(t) + \sum_{j=1}^{2}\sum_{k=1}^{2} d_{1jk}f_j(x_j(t)) + q_1, \\ \\ D^{\omega(\alpha)}x_2(t) = -p_2 x_2(t) + \sum_{j=1}^{2}\sum_{k=1}^{2} d_{2jk}f_j(x_j(t)) + q_2, \end{cases} \tag{54}$$

*where* $\omega(\alpha) = \tau_1\delta(\alpha - \alpha_1) + \tau_2\delta(\alpha - \alpha_2)$, $\alpha_1 = \frac{2}{3}, \alpha_2 = \frac{1}{3}$, $\delta(\cdot)$ *denotes Dirac function;* $f_j(x_j(x)) = \tanh(x)$, $j = 1, 2; q_i = 1$, $i = 1, 2$.

$$P = diag(p_1, p_2) = \begin{pmatrix} 1.5 & 0 \\ 0 & 1 \end{pmatrix},$$

$$D_1 = (d_{ij1})_{2\times2} = \begin{pmatrix} 1 & 2 \\ 2 & 1 \end{pmatrix},$$

$$D_2 = (d_{ij2})_{2\times2} = \begin{pmatrix} 2 & 1 \\ 1 & 2 \end{pmatrix},$$

*The response system is depicted as*

$$
\begin{cases}
D^{\omega(\alpha)} y_1(t) = -p_1 y_1(t) + \sum\limits_{j=1}^{2} \sum\limits_{k=1}^{2} d_{1jk} f_j(y_j(t)) + q_1 + u_1(t), \\[2ex]
D^{\omega(\alpha)} y_2(t) = -p_2 y_2(t) + \sum\limits_{j=1}^{2} \sum\limits_{k=1}^{2} d_{2jk} f_j(y_j(t)) + q_2 + u_2(t),
\end{cases}
\tag{55}
$$

*where $\omega(\alpha), P, D_i, q_i, \ i = 1, 2$ are the same as the drive system (54).*

*Take $\beta_1 = 1.5, \beta_2 = 1, g_1 = 2, g_2 = 1.5, \tau_1 = 1, \tau_2 = 2$, by calculation, we have$\sum_{i=1}^{n} \sum_{j=1}^{m}$ $\sum_{k=1}^{n} \left[ 2(p_i + g_i) - \mu d_{ijk} F_j - \frac{1}{\mu} d_{jik} F_i \right] = 1 > 0$ and the error bounded $\sqrt{\frac{1}{\lambda_5 \min_{1 \le i \le n} \xi_i}} = 1$, by similar with the processes in Example 1, we can obtain that all conditions in Theorem 3 hold. Hence, it follows from Theorem 3 that systems (54) and (55) can achieve quasi-projectively synchronized. Figure 3 is figure of numerical simulation result.*

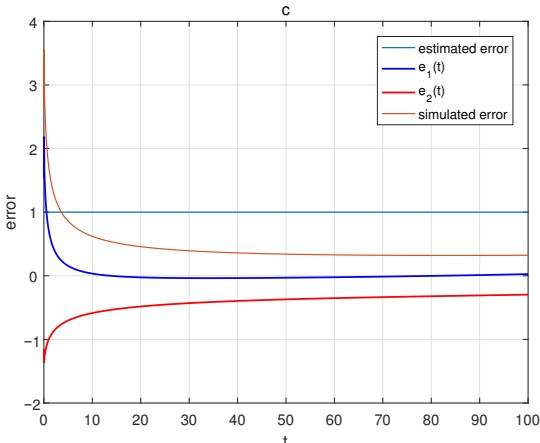

**Figure 3.** The state of error system when $\beta_1 = 1.5, \beta_2 = 1, g_1 = 2, g_2 = 1.5$.

## 5. Conclusions

In this paper, the problems of quasi-projective synchronization of distributed-order recurrent neural networks have been investigated. Based on the definition of the distributive-order derivative and the metric space theory, some lemmas that are widely used in fractional order systems have been generalized to distributed-order systems. Afterwards, according to the Lyapunov method, Laplace transform, Laplace final value theorem, and some inequality techniques, the quasi-projective synchronization of the aforementioned neural networks have been investigated. Finally, two numerical examples have been given to manifest the validity of the theoretical results.

We would like to point out that it is possible to use our method to discuss some dynamic behavior, such as synchronization, passivity and state estimation of distributed-order complex-valued neural networks and quaternion-valued neural networks. The corresponding results will be carried out in the near future.

**Author Contributions:** Conceptualization, Writing—original draft, X.L.; Project administration, Writing—review & editing, K.L.; Supervision, Writing—review & editing, Q.S.; Software, Revised manuscript, X.Y. All authors have read and agreed to the published version of the manuscript.

**Funding:** This research received no external funding.

**Institutional Review Board Statement:** Not applicable.

**Informed Consent Statement:** Not applicable.

**Data Availability Statement:** Not applicable.

**Acknowledgments:** This work was supported by the National Natural Science Foundation of China under Grant 61573010 and 61906023, the Opening Project of Sichuan Province University Key Laboratory of Bridge Non-Destruction Detecting and Engineering Computing under Grant 2021QYJ06, and in part by the Chongqing Research Program of Basic Research and Frontier Technology under Grant cstc2019jcyj-msxmX0710.

**Conflicts of Interest:** The authors declare no conflict of interest.

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
