# Peer review of "Quasi-Projective Synchronization of Distributed-Order Recurrent Neural Networks"

_fractalfract, doi:10.3390/fractalfract5040260_

Round 1

Reviewer 1 Report

This paper investigates  the quasi-projective synchronization of
distributed-order recurrent neural networks and by using Lyapunov
method, Laplace transform, Laplace final value theorem, and some
inequality techniques, obtained some sufficient on the
quasi-projective synchronization for the considered system. Finally,
two numerical examples are given to verify the effectiveness of the
theoretical results. The techniques and results of this paper are
novel and interesting, however, there still exist some problems to
be addressed.(See attachment)

In conclusion, I would suggest acceptance with minor revision.

Reviewer 2 Report

  1. Introduction section should be presented such that justification of problem statements become more evident and contribution of the proposed work is further highlighted. Additionally, it is suggested to segment the introduction into three separate sub-sections i.e., (1 Introduction, 1.1 related work, 1.2 Innovative insights and contributions, 1.3 organization. Salient features should also be listed in bullets form in the innovation contribution sub-section.
  2. The Process flow diagram, and Pseudocode of proposed design methodology should be provided in the body of the manuscript with necesary mathematics as well as details of the steps.
  3. Results and discussion section should be presented with few more graphical and numerical illustration in order to analyze the performance more elaboratively. Statistical observations on the performance should be given by using different forms of parametric and non-parametric tests.
  4. Potential applications of proposed methodology in diversified fields should be given in the conclusion section as a future related works.

The proposed methodology can be a good alterantive for problems in astrophysics, plasma physics, atomic physics, thermodynamics, electromagnetic, machines, nanotechnology, fluid mechanics, electrohydrodynamics, signal processing, power, energy, bioinformatics, economic and finance. Please see the relevant reference on internet.

Reviewer 3 Report

Kindly find the attachment.

Round 2

Reviewer 2 Report

Revised version improve so recommended for acceptance

Reviewer 3 Report

The revised version is well rewritten and respond to all comments correctly. In my view, thus, it can be accepted for publication.

Also, the authors should concentrate more on the typo errors, English correction and grammar mistakes once again throughout the manuscript.

For example: Fig.3 is figure of numerical simulation result.